# Learning Combinatorial Node Labeling Algorithms

## Abstract

We present the combinatorial node labeling framework, which generalizes many prior approaches to solving hard graph optimization problems by supporting problems where solutions consist of arbitrarily many node labels, such as graph coloring. We then introduce a neural network architecture to implement this framework. Our architecture builds on a graph attention network with several inductive biases to improve solution quality and is trained using policy gradient reinforcement learning. We demonstrate our approach on both graph coloring and minimum vertex cover. Our learned heuristics match or outperform classical hand-crafted greedy heuristics and machine learning approaches while taking only seconds on large graphs. We conduct a detailed analysis of the learned heuristics and architecture choices and show that they successfully adapt to different graph structures.

## 1 Introduction

Graph problems have numerous real-world applications, ranging from scheduling problems (Marx, 2004) and register allocation (Chaitin, 1982; Smith et al., 2004), to computational biology (Abukhzam et al., 2004). However, many useful graph optimizations problems are NP-hard to solve (Karp, 1972). This has spurred a variety of approaches, from greedy heuristics (Brélaz, 1979; Papadimitriou & Steiglitz, 1982; Matula & Beck, 1983; Avis & Imamura, 2007; Delbot & Laforest, 2008) to integer linear programming (Graver, 1975). More recently, machine learning approaches have shown increasing promise (Dai et al., 2017; Kool et al., 2019; Li et al., 2018; Karalias & Loukas, 2020).

From a structural point of view, many graph problems fall into one of three classes depending on the type of their solution: Problems that ask for (1) subsets of vertices, (2) permutations of vertices, or (3) partitions of vertices into two or more sets. Most work has focussed on either the first two (Dai et al., 2017), or just one of the three (Bello et al., 2017; Li et al., 2018; Kool et al., 2019; Karalias & Loukas, 2020; Manchanda et al., 2020; Cappart et al., 2020; Drori et al., 2020; Ma et al., 2020). Existing machine learning methods for the first two types of problems, such as S2V-DQN (Dai et al., 2017), do not easily generalize to cases where the number of labels is not known in advance. Many important and challenging problems, such as graph coloring (Marx, 2004; Myszkowski, 2008; Bandh et al., 2009), require that vertices be partitioned into *an unkown number of sets*.

To address this, we present the *combinatorial node labeling* framework (§2), which generalizes prior approaches (Fig. 1), and supports many problems, including minimum vertex cover (Onak et al., 2012; Bhattacharya et al., 2017; Ghaffari et al., 2020), traveling salesman (Dantzig et al., 1954; Garey & Johnson, 1990), maximum cut (Karp, 1972), and list coloring (Jensen et al., 1995). These, and many other (§D), problems can all be framed as iteratively assigning a *label* to nodes, in some order. We then introduce a neural architecture, GAT-CNL, to learn greedy-inspired heuristics for such problems (§3). We use policy gradient reinforcement learning (Sutton & Barto, 2018; Kool et al., 2019) to learn a node ordering and combine this with a fixed label rule to label each node according to the ordering. We show that for the chosen label rules, there still exists an order that guarantees an optimal solution. By using policy gradients, we can construct both a deterministic greedy policy, as well as a probabilistic policy where sampling boosts the solution quality. To improve performance, we incorporate two inductive biases: *spatial locality*, where labeling a node only impacts the weights of its neighbors; and *temporal locality*, where node selection is conditioned only on the previously labeled node, a summary of prior labelings, and a global graph context (Figs. 2 and 3).

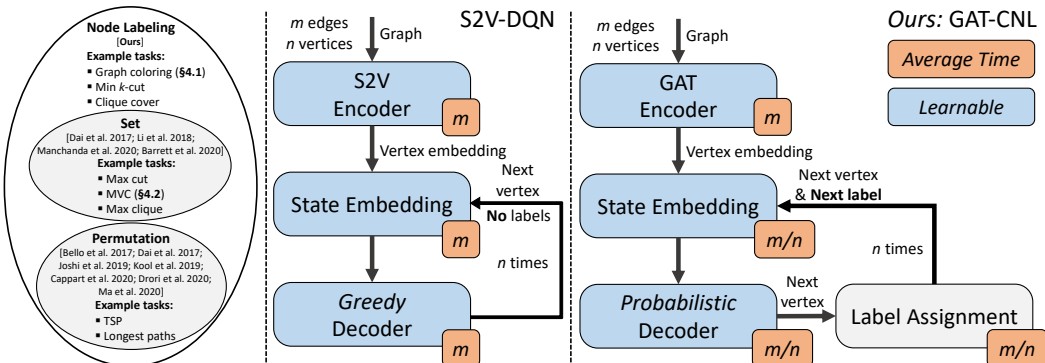

Figure 1: **Left:** Venn diagram of tasks solvable with the set, permutation, and node labeling frameworks. Node labeling generalizes existing frameworks and allows solving additional tasks. **Center & right:** Comparison of our architecture with S2V-DQN (Dai et al., 2017). We add a *label assignment* step, allowing us to solve new problems. Further, the average time for picking the next vertex is significantly reduced, such that the total number of arithmetic operations is now linear in the size of the graph.

We evaluate our approach (§4) and demonstrate significantly improved performance for neural graph coloring (GC) and find near-optimal solutions for minimum vertex cover (MVC). We additionally study the runtime of our models, conduct comprehensive ablation studies, and provide qualitative analyses of the learned heuristics, showing they adapt to the properties of the input graph.

**Related work.** We now review key related works. Figure 1 (left) provides a comparison of node labeling with other frameworks.

**Supervised learning.** The fundamental downsides of supervised learning for combinatorial optimization are twofold: First, it can be difficult to formulate a problem in a supervised manner, since it might have many optimal solutions (e.g., GC). Second, even if the problem admits a direct supervised formulation, we still need labeled data for training, which can be hard to generate and relies on an existing solver. In particular, supervised learning cannot easily be used for problems that have not been studied before. Advantages of supervised learning are its sample efficiency and that it can lead to overall better results. Recent approaches like Joshi et al. (2019) and Manchanda et al. (2020) obtain good results for influence maximization (IM) and the traveling salesman problem (TSP), respectively. Both approaches use supervised learning. For IM, the approach of Manchanda et al. (2020) shows promising results on graphs much larger than those seen in training. For TSP, the approach of Joshi et al. (2019) is very efficient but does not generalize well to graphs larger than those seen in training. Li et al. (2018) also use supervised learning and produce good results on minimum vertex cover (MVC), maximum independent set, and maximal clique.

**Unsupervised Learning.** To apply unsupervised learning, it is necessary to formulate a differentiable surrogate loss. There have been approaches for several specific combinatorial optimization problems (Nazi et al., 2019; Amizadeh et al., 2019; Tönshoff et al., 2019; Yao et al., 2019) and there has been progress to create a framework for the derivation of trainable losses (Karalias & Loukas, 2020). Still, significant insight into a problem is required to design suitable loss functions.

**Reinforcement learning** (RL)**.** Using RL only requires a way to represent partial solutions and a way to score the cost of a (partial or final) solution. Dai et al. (2017) provide S2V-DQN, a general framework for learning problems like MVC and TSP that is trained with RL. It shows good results across different graph sizes for the covered problems, but is not fast enough to replace existing approaches nor does it handle arbitrary node labels (see Fig. 1). Kool et al. (2019) focus on routing problems like TSP and the vehicle routing problem. They outperform Dai et al. on TSP instances of the training size. Unfortunately, their approach does not seem to generalize to graph sizes that are very different from those used for training. Several other RL approaches have been proposed and evaluated for TSP (Bello et al., 2017; Cappart et al., 2020; Drori et al., 2020; Ma et al., 2020). Barrett et al. (2020) consider the maximum cut (MaxCut) problem. Huang et al. (2019) present a Monte Carlo search tree approach specialized only for graph coloring. These methods do not address the general node labeling framework, but instead model the solution as a permutation of vertices (e.g., TSP, vehicle routing) or a set of nodes or edges (e.g., MVC, MaxCut). Instead, we can represent

solutions where vertices are assigned to an unknown and unbounded number of partitions, which is crucial for solving tasks such as graph coloring.

## 2 COMBINATORIAL NODE LABELING

Many graph heuristics can be phrased as a greedy process, where vertices get assigned a problem-dependent *label* one after the other. For example, this label could indicate if the vertex is part of the solution set, its position in a permutation, or its membership in one of many sets. We introduce *combinatorial node labeling*, which frames many hard graph optimization problems, such as graph coloring (see §D for a list), as a greedy process. This generalizes previous work (Kool et al., 2019; Dai et al., 2017; Ma et al., 2020; Drori et al., 2020), to encompass problems where the number of labels is not known in advance and is unbounded (see Fig. 1).

Every node labeling problem can be formulated as a (finite) Markov decision process (MDP), during which nodes are successively added to a so-called *partial* node labeling until a termination criterion is met. In §3, we will present a graph learning approach to optimizing such node labeling MDPs.

### 2.1 PRELIMINARIES

We consider an undirected, unweighted, and simple graph $G = (V, E)$ with $n$ nodes in $V$ and $m$ edges in $E$. We denote the neighbors of a node $v$ by $N(v)$. We assume w.l.o.g. that the graph is connected and hence $m = \Omega(n)$.

A *node labeling* is a function $c : V \to \mathcal{L}$, where $\mathcal{L} \subseteq \{0, \dots, n\}$. A *partial node labeling* is a function $c' : V' \to \mathcal{L}'$ for a subset of nodes $V' \subseteq V$ and labels $\mathcal{L}' \subseteq \mathcal{L}$. A node labeling problem is subject to a feasibility condition and a real-valued *cost function* $f$. The cost function maps a node labeling $c$ to a real-valued cost $f(c)$. We require that the feasibility condition be expressed in terms of an efficient (polynomial-time computable) *extensibility test* $T : \mathcal{P}(V \times \mathcal{L}) \times V \times \mathcal{L} \to \{0, 1\}$, where $\mathcal{P}$ denotes the powerset. We say the extensibility test passes when it returns 1.

Intuitively, given a partial node labeling $c'$, a node $v \notin V'$, and label $\ell$, the extensibility test passes if and only if $c'$ can be extended by labeling node $v$ with $\ell$ such that $c'$ can be extended into a node labeling. Formally, the extensibility test characterizes the set of *feasible solutions*:

**Definition 2.1.** A node labeling $c$ is feasible if and only if there exists a sequence of node-label pairs $(v_1, \ell_1), \dots, (v_n, \ell_n)$ such that for all $i \geq 0$ the extensibility test $T$ satisfies $T(\{(v_1, \ell_1), \dots, (v_i, \ell_i)\}, v_{i+1}, \ell_{i+1}) = 1$.

The goal of the node labeling problem is to *minimize* the value of the cost function among the feasible node labelings. For consistency, an infeasible node labeling has infinite cost. Next, we present the two node labeling problems on which we focus in our evaluation.

**Definition 2.2.** A $k$-coloring of a graph $G = (V, E)$ is a node labeling $c : V \to \{1, 2, \dots, k\}$ such that no two neighbors have the same label, i.e., $\forall \{u, v\} \in E : c(u) \neq c(v)$.

The cost function for GC is the number of distinct labels (or colors) $k$. Given a partial node labeling $c' : V' \to \{1, \dots, k\}$ and any vertex-label pair $(v, \ell)$, the extensibility test passes for $(c', v, \ell)$ if and only if the extended partial node labeling $c' \cup (v, \ell)$ is a $k$- or $(k+1)$-coloring of the induced subgraph $G[V' \cup \{v\}]$. In particular, the test does not pass when $\ell > k + 1$. The smallest $k$ for which there is a $k$-coloring of $G$ is the *chromatic number* $\chi(G)$ of $G$.

**Definition 2.3.** A vertex cover of a graph $G = (V, E)$ is a node labeling $c : V \to \{0, 1\}$ such that every edge is incident to at least one node with label 1, i.e., $\forall \{u, v\} \in E : c(u) = 1 \lor c(v) = 1$.

The cost function for MVC is the number of nodes with label 1. Given a partial node labeling $c' : V' \to \{0, 1\}$ the extensibility test passes for $(c', v, \ell)$ if and only if the extended partial node labeling $c' \cup (v, \ell)$ is a vertex cover of the induced subgraph $G[V' \cup \{v\}]$.

### 2.2 NODE LABELING MDP

We show how to construct an MDP that models a given combinatorial node labeling problem. Minimizing the cost of the combinatorial node labeling problem is equivalent to maximizing the return of this MDP. In the vast majority of reinforcement learning approaches to solve combinatorial

graph optimization problems (Kool et al., 2019; Dai et al., 2017; Ma et al., 2020; Drori et al., 2020), a state corresponds to a set or sequence of nodes that are already added to a solution set. Instead, in our setting the state represents a partial node labeling. This is why in addition to problems like MVC and TSP, *we can also model problems with more than two labels* (even when the number of labels is not known in advance). Graph coloring is such a problem.

**Lemma 2.4.** *For any node labeling problem, there is an MDP whose terminal states correspond to the feasible solutions with a cost equal to the negative return.*

We embed the cost function $f$ and the extensibility test into the MDP. Note that we do not require a way to measure the cost of partial node labelings. Here, we formulate the state space, action space, transition function, and reward. In §C.1, we finish the proof of Lemma 2.4.

**State space.** A state $S$ represents a partial node labeling. It is a set of pairs $S = V' \times \mathcal{L}$ for a subset of nodes $V' \subseteq V$ and a subset of labels $\mathcal{L} \subseteq \{0, \dots, n\}$. A state is terminal if $V' = V$. Hence, the set of states is the powerset $\mathcal{P}(V \times \{0, \dots, n\})$ of the Cartesian product of the vertices and labels.
**Action space.** In state $S$, the set of legal actions are the pairs $(v, \ell)$ for nodes $v$ and labels $\ell$ which pass the extensibility test of the problem for the partial node labeling given by $S$ (i.e., $T(S, v, l) = 1$).
**Transition function.** In our case, the transition function $\mathcal{T}$ is deterministic. That is, given the current state $S_t$ and an action $(v, \ell)$, $\mathcal{T}(S_t, (v, \ell))$ yields the next state $S_{t+1} = S_t \cup \{(v, \ell)\}$.
**Reward**. For a terminal state $S$ representing the node labeling $c$, the reward is $-f(c)$. For all other states, the reward is $0$.

A *policy* is a mapping from states to probabilities for each action. Note that we can turn a probabilistic policy into a deterministic *greedy* policy by choosing the action with largest probability. Next, we present how to train such a policy *end-to-end* using policy gradients.

## 3 GRAPH LEARNING APPROACH

We present a graph learning approach to node labeling, which is inspired by greedy algorithms. Greedy approaches generally trade optimality for improved runtime. A greedy node labeling algorithm assigns a label in $\{0, \dots, n\}$ to one node after another based on a problem-specific heuristic. Hence, it can be seen as providing (1) an order on the nodes and (2) a rule to label the next selected node.

We focus on learning an order on the nodes and pick a label that passes the extensibility test according to a fixed rule. The following two lemmas show there exists a *label assignment rule* that ensures the optimal solution can be found for GC and MVC (see §C.2 for the proofs):

**Lemma 3.1.** *For every graph $G$, there exists an ordering of vertices for which choosing the smallest color that passes the extensibility test colors $G$ optimally.*

**Lemma 3.2.** *For every graph $G$, there exists an ordering of vertices for which choosing the label $1$ until every edge in $G$ has one of its endpoints labeled with $1$ produces a minimum vertex cover of $G$.*

We expect similar results can be obtained for most other node labeling problems.

Instead of a handcrafted ordering heuristic, we learn to assign weights to each node and choose the nodes according to their weights. To compute these weights, we introduce a novel *spatial locality* inductive bias inspired by the greedy heuristics: labeling a node should only affect the weights of its neighbors. As we will show in §4.3, this leads to better test scores compared to the alternatives of updating all or none of the weights when a node is labeled. This spatial locality bias is inspired by successful greedy heuristics: The ListRight heuristic for MVC (Delbot & Laforest, 2008) assigns a node to the vertex cover based on the assignment of its neighbors. For GC, the DSATUR strategy selects nodes according to their saturation degree (Brélaz, 1979). If a new node is selected, only the saturation degree of its neighborhood can change; the others remain unchanged.

### 3.1 POLICY OPTIMIZATION

We train our node labeling model by policy gradients, specifically REINFORCE (Bello et al., 2017) with a greedy rollout baseline (Kool et al., 2019). The advantage of policy gradients over Q-learning is that is has stronger convergence guarantees (Sutton & Barto, 2018). At a high level, the algorithm works as follows. We begin by initializing two models, the *current* model and the *baseline* model. For each graph in the batch, the algorithm performs a probabilistic rollout of the policy. The baseline

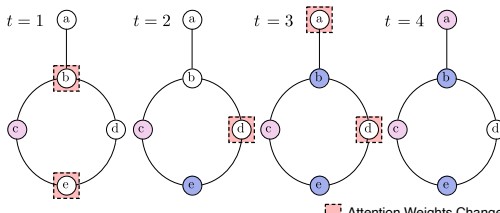

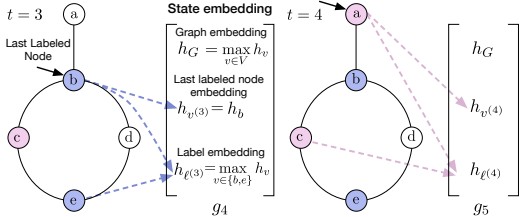

Figure 2: *Spatial locality of the decoding.* We show how a graph is 2-colored using node order $c, e, b, a, d$. After labeling a node, only its neighbors' attention weights change; e.g., when $c$ is colored, only its neighbors $b$ and $e$ receive new attention weights. (We omit the last step where $d$ is colored.)

Figure 3: *Temporal locality of the state embedding.* We show how the state embedding is updated as nodes are colored. The state embedding focuses on the last labeled node, and contains the graph embedding, and embeddings of the last labeled node and its label, which pools the embeddings of nodes with the same label.

model performs a greedy rollout. The difference between the two costs determines the policy gradient update. After every epoch, we perform a (one-sided) paired $t$-test over the cost on a challenge dataset to check if the baseline model should be replaced with the current model. See §A.2 for more details.

## 3.2 GAT-CNL ARCHITECTURE

Our architecture, GAT-CNL, consists of an encoder and a decoder to learn a policy specific to the node labeling problem. The encoder learns the local structural information that is important for the problem in the form of a node embedding.

The *state embedding* encapsulates information about the graph itself (enabling the network to adapt its actions to the graph), the last node that was labeled and its label, and a summary of prior actions (with pooling). This enables the state embedding to have constant size; adding additional nodes provided no benefit (see §4.3). This also serves as a *temporal locality* bias; however, note that the decoder is *not* Markovian, as it depends on more than just the previous decision.

The decoder uses the node embeddings and the state embedding to select the next node based on attention weights between the node embeddings and the state embedding. After the decoder picks the next node $v$, the label rule (see Lemmas 3.1 and 3.2) assigns the label $\ell$ for the node. The policy then takes the action $(v, \ell)$. Then, the state embedding is updated and the decoder is invoked again until all nodes are labeled. Figure 1 (right) overviews our architecture.

**Node features.** Each node $v$ is associated with an input feature vector $x_v$. Our input features consist of a combination of sine and cosine functions of the node degree, similar to positional embeddings (Vaswani et al., 2017). This representation ensures that input features are bounded in magnitude even for larger graphs. We subtract the mean node degree from the degrees on the synthetic dense graph instances.

**GAT encoder.** We use a hidden dimension of size $d$ (unless stated otherwise, $d = 64$). The input features are first linearly transformed and then fed into a GNN, which produces, for each node $v$, a node embedding $h_v \in \mathbb{R}^d$. We use a three-layer Graph Attention Network (GAT) (Vaswani et al., 2017; Velickovic et al., 2018; Lee et al., 2019), additive multi-head attention with four heads, batch normalization (Ioffe & Szegedy, 2015) with a skip connection (He et al., 2016) at each encoder layer, and leaky ReLU activations (Maas et al., 2013).

**State embedding.** The *state embedding* allows the decoder to condition its choice based on the graph instance and the partial node labeling. For computational reasons, we ensure it is of constant size. Denote the node labeled in step $t$ by $v^{(t)}$ and its label by $\ell^{(t)}$. Then the state embedding consists of three components concatenated together: (1) The *graph embedding* $h_G$, a max-pooling over all node embeddings. (2) The node embedding $h_{v^{(t-1)}}$ of the *last labeled node* $v^{(t-1)}$. (3) The label embedding $h_{\ell^{(t-1)}}$ of the last labeled node's label $\ell^{(t-1)}$, a max-pooling over the embeddings of all nodes with that label. In the first iteration, we use a learned parameter $h^{(0)}$ for (2) and (3). We considered including more than just the last labeled node, but found that this led to worse performance (§4.3). Hence, this induces a *temporal bias* by focusing on the prior node and nodes with the same label as the last labeled node. See Figure 3 for an illustration of the state embedding.

**Local attention decoder.** The decoder takes as input the node embeddings generated by the encoder and the state embedding and outputs the next node to label. In each time step $t$, an attention mechanism between the state embedding $g_t$ and each node embedding $h_v$ produces attention weights $a_v^{(t)}$. Here, we introduce a *spatial locality* bias: labeling a node can only affect the attention scores of its neighbors in the next time step. Let $V'$ be the set of nodes already labeled. The attention weight $a_v^{(t)}$ for node $v$ in time step $t$ is given by the *local decoding*. For a node $v \notin V'$:

$$a_v^{(t)} = \begin{cases} C \cdot \tanh\left( \frac{(\Theta_1 g_t)^T (\Theta_2 h_v)}{\sqrt{d}} \right) & v \in \mathcal{N}(v^{(t-1)}) \text{ or } t = 0 \\ a_v^{(t-1)} & v \notin \mathcal{N}(v^{(t-1)}). \end{cases}$$

If $v \in V'$, then the attention weight is $a_v^{(t)} = -\infty$. In the first iteration of the decoder, we calculate the coefficients for each node in the graph. As in Bello et al. (2017), we clip the attention coefficients within a constant range $[-C, C]$. In our experiments we set $C = 10$. The learnable parameter matrices are $\Theta_1 \in \mathbb{R}^{d \times 3d}$ and $\Theta_2 \in \mathbb{R}^{d \times d}$. We use scaled dot-product attention (Vaswani et al., 2017) (instead of additive attention) to speed up the decoding. Finally, for each node $v$ we apply a softmax over all attention weights to obtain the probability $p_v$ that node $v$ is labeled next. See Figure 2 for a visualization of the attention weight computation during decoding.

During inference, our *greedy policy* selects the vertices with maximum probability. Our *sampling policy* (for $k$ samples) runs the greedy policy once, then evaluates the learned probabilistic policy $k$ times (selecting a node $v$ with the learned probability $p_v$), returning the best result.

### 3.3 NUMBER OF OPERATIONS

We express the number of operations (arithmetic operations and comparisons) of the model during inference parameterized by the embedding dimension $d$, the number of nodes $n$ and the number of edges $m$. The encoder uses $\mathcal{O}(dm + d^2 n)$ arithmetic operations and the decoder uses $\mathcal{O}(d^2 m)$ arithmetic operations, resulting in $\mathcal{O}(dm + d^2 n + d^2 m)$ arithmetic operations, which is *linear in the size of the graph*. To select the action of maximum probability (or sample a node), the decoder additionally needs $\mathcal{O}(n^2)$ comparison operations (although this could be reduced to $\mathcal{O}(m \log n)$ with an appropriate data structure). We empirically study the runtime in §§4.1 and 4.2; in practice, the $d^2 m$ term dominates the runtime for the evaluated graphs until 5000 vertices. In contrast, updating all attention weights after every labeling scales as $\mathcal{O}(n^3)$ (see §B.5).

## 4 EXPERIMENTS

We evaluate our approach on established benchmarks for graph coloring and minimum vertex cover, including greedy baselines and machine learning approaches. We focus on other heuristic approaches that return an approximation in polynomial time.

**Training.** We use three different synthetic graph distributions to generate instances for training and validation (Albert & Barabási, 2002; Erdős & Rényi, 1960; Watts & Strogatz, 1998). We generate 20,000 graphs for training. The graphs have between 20 and 100 nodes. We use Adam with learning rate $\alpha = 10^{-4}$ (Kingma & Ba, 2015). The effective batch size is $B = 320$, which comes from using batches of 64 graphs for each node count $n$ and accumulating their gradients. We clip the L2 norm of the gradient to 1, as done in Bello et al. (2017). We selected these hyperparameters after initial experiments on the validation set. Each model took 15–20 CPU compute node hours to train on a cluster with Intel Xeon E5-2695 v4 and 64 GB memory per node. The overall time spent training was

Table 1: Graph coloring results on the Lemos et al. (2019) subset of the COLOR challenge graphs.

| | Method | Cost | Wins | Optimal |
|---|---|---|---|---|
| Classic | Largest First | 10.65 | 50% | 45% |
| | DSATUR | 9.85 | 65% | **50%** |
| | Smallest Last | 10.8 | 50% | 45% |
| ML | Lemos et al. (2019) | N/A | 45% | 25% |
| | Ours — Greedy | $10.36^{\pm 0.01}$ | 55% | 50% |
| | Ours — Sampling | $\mathbf{9.65}^{\pm 0.04}$ | **70%** | **50%** |

Table 2: Comparison of MVC approaches on dense ER graphs with edge-probability 0.15.

| | Method | Cost | Approx. Ratio |
|---|---|---|---|
| Class. | Maximal Matching | 232.00 | 1.2486 |
| | List Right | 225.35 | 1.1120 |
| ML | Li et al. (2018) | **212.296** | 1.0594 |
| | S2V-DQN | N/A | 1.1208 |
| | Ours — Greedy | $221.52^{\pm 1.1}$ | 1.0510 |
| | Ours — 10 samples | $220.27^{\pm 1.2}$ | **1.0443** |

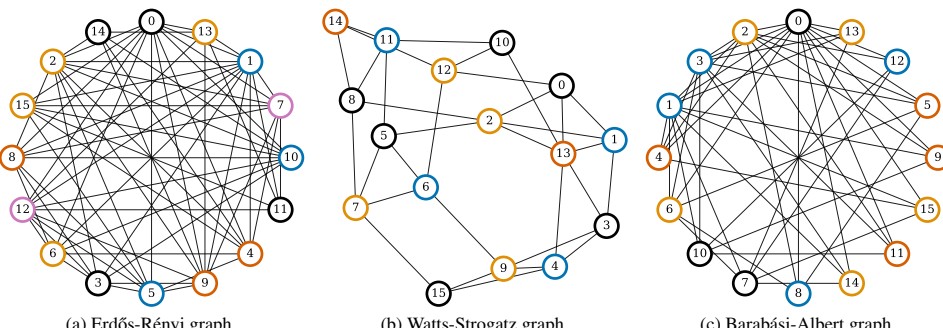

| (a) Erdős-Rényi graph. | (b) Watts-Strogatz graph. | (c) Barabási-Albert graph. |

Figure 4: Example colorings produced by our learned heuristic. Node borders indicate the colors. Numbers on the nodes indicate the order in which the heuristic labels them.

was less than 2500 CPU node hours and the time spent on validation and testing was less than 300 CPU node hours. We train each model for 200 epochs with five random seeds and report the standard deviation $\sigma$ of cost w.r.t. the random seeds as $\pm\sigma$. See §A for more details.

**Test Scores.** In addition to mean cost, we report the ratio of the solution cost to the optimal solution cost (*approximation ratio*). For large graphs, this cannot be computed exactly in a timely manner. In this case, we use the best solution found by an ILP solver within a compute time of one hour. To compare with baselines which return infeasible solutions (and hence have ill-defined cost), we report the percentage of *wins* (ties for first place count as wins) and the percentage of instances solved *optimally*. We refer to these metrics as "Wins" and "Optimal", respectively. We use the model with the lowest cost to compute these percentages.

**Runtime.** We benchmark on a c2d-standard-4 Google Cloud instance with 4 vCPUs and 16 GB RAM.

### 4.1 GRAPH COLORING

**Greedy baselines.** *Largest-First* greedily colors nodes in decreasing order of degree. *Smallest-Last* (Matula & Beck, 1983) colors the nodes in reverse degeneracy order, which guarantees that when a node is colored, it will have the smallest possible number of neighbors that have been already colored. Smallest-Last guantees a constant number of colors for certain families of graphs, such as Barabási-Albert graphs (Albert & Barabási, 2002) and planar graphs (Matula & Beck, 1983). *DSATUR* (Brélaz, 1979) selects nodes based on the largest number of distinct colors in its neighborhood. DSATUR is exact on certain families of graphs, e.g., bipartite graphs (Brélaz, 1979). We use the implementations from NetworkX (Hagberg et al., 2008).

**Machine learning baseline.** We compare our approach with the chromatic number estimator of Lemos et al. (2019). It does not guarantee that the solution is feasible, meaning that it can both under- and overestimate the chromatic number. We use the values reported by the original paper. Note that S2V-DQN (Dai et al., 2017) cannot solve GC because of the way it embeds the state.

**COLOR benchmark** (Table 1). We evaluate on the same subset of the COLOR02/03 benchmark (Col, 2002) as Lemos et al. (2019), consisting of 20 instances of size between 25 and 561 nodes. Our greedy policy outperforms both Largest-First and Smallest-Last and is tied with DSATUR for the most graphs solved optimally. When sampling (100 samples) to evaluate the policy, our model outperforms all baselines in *both mean cost and win percentage* and is tied for the most graphs solved optimally. The approximation ratio is 1.25 and 1.13 for our greedy and sampling policies, resp.

**Results on classic graphs.** We also trained our model on four families of sparse graphs: cycles, wheels, random trees, and stars. We trained on graphs up to 400 nodes and evaluated on graphs up to 10,000 nodes. The produced colorings are optimal or extremely close to optimal for all four families (Table 3). As our model works perfectly on cycles and wheels we conclude that the model learns to leverage local graph structure and works even when all nodes have the same degree and are completely symmetrical. Table 4 shows the validation cost for varying instance size on Watts-Strogatz graphs, which grows only slowly with instance size. See §B.2 for additional results.

**Qualitative results.** Figure 4 presents typical examples of the learned coloring heuristic on the training distribution graphs. See §E.1 for more examples. We can observe that the heuristic generally

Table 3: Our approach colors simple families of graphs (near-)optimally.

| Graph Family | | Optimal $\chi$ | Ours |
|---|---|---|---|
| Stars | ✳ | 2 | $\mathbf{2}^{\pm 0.0}$ |
| Random trees | ↜ | 2 | $\mathbf{2}^{\pm 0.01}$ |
| Even cycles | ⬠ | 2 | $\mathbf{2}^{\pm 0.0}$ |
| Odd cycles | ⟳ | 3 | $\mathbf{3}^{\pm 0.0}$ |
| Odd wheels | ⊠ | 3 | $\mathbf{3}^{\pm 0.01}$ |
| Even wheels | ✵ | 4 | $\mathbf{4}^{\pm 0.0}$ |

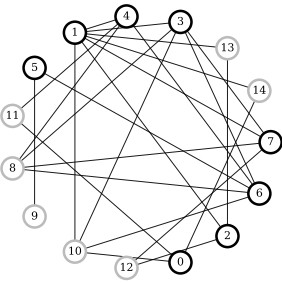
(a) Erdős-Rényi graph.

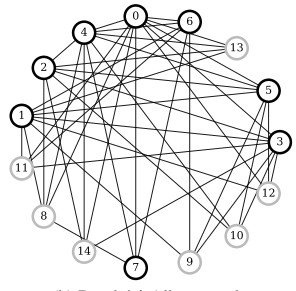
(b) Barabási-Albert graph.

Figure 5: Example covers from our learned heuristic. Nodes with a bold border are in the cover. Numbers indicate the labeling order. Once a cover is found, the order is irrelevant.

picks higher degree, centrally located nodes first. However, if several nodes have the same degree, it favors coloring neighboring nodes subsequently. This happens in the WS graphs, see Figure 4b. The learned heuristic can consistently color the WS graphs with 4 colors, which matches the Smallest-Last heuristic. We conclude that the learned heuristic captures complex aspects of the graph extending beyond simple degree-based decisions and considers the graph's local neighborhood structure.

**Runtime** (Figure 6). We compare the runtime of our approach with the classical baselines. As we did not have access to the code of Lemos et al. (2019), we could not compare the runtime directly. Our approach is faster than DSATUR for graphs larger than 640 nodes and scales much better. As expected from §3.3, the runtime of our algorithm grows linearly with the size of the graph, similar to the simpler baselines such as Largest First and Smallest Last, which have better constant factors.

## 4.2 MINIMUM VERTEX COVER

**Classic baselines.** We compare with two classic algorithms. First, we use the endpoints of a maximal matching, which produces a 2-approximation Papadimitriou & Steiglitz (1982). Second, we compare with list-right Delbot & Laforest (2008), a $\frac{\sqrt{\Delta}}{2} + \frac{3}{2}$ approximation algorithm for maximum degree $\Delta$.

**Machine learning baselines.** *S2V-DQN* is a $Q$-learning based approach (Dai et al., 2017). We use the values reported in the original paper. Li et al. (2018) present a tree-search based approach trained in a supervised way. In contrast to S2V-DQN, it samples multiple solutions, then verifies if they are feasible. The time to construct a feasible solution varies depending on the instance. We use the publicly available code and pretrained model from the authors. We run Li et al.'s code until it finds a feasible solution, and sample more solutions if it is below the time budget of 30 seconds per graph.

**Results on in-distribution graphs.** We evaluate and compare our approach for MVC with S2V-DQN (Dai et al., 2017) and Li et al. (2018) on the same dataset of generated graphs as Dai et al. (2017). It consists of 16,000 graphs from two distributions, Erdős-Rényi (ER) (Erdős & Rényi, 1960) and Barabási-Albert (BA) (Albert & Barabási, 2002), of sizes varying from 20 to 600 nodes. We use the results reported by Dai et al. (2017) on their model trained on 40–50 nodes, except for the graphs with less than 40 nodes, for which no data is available for this model. Hence we use their model trained on 20–40 nodes on these smaller graphs. See Table 2 for the results on ER graphs and §B.3 for results on additional graphs. On ER graphs, our model achieves the closest average approximation ratio, followed by Li et al. (2018). Li et al. (2018) has the lowest average cost. Note that the lowest approximation ratio and lowest cost need not coincide because the cost grows quickly with graph size, whereas the approximation ratio does not. Our model and Li et al. (2018) outperform the greedy baseline, while S2V-DQN is slightly outperformed by List Right. In Table 4, we show how the approximation ratio and cost depend on the instance size. As shown in §B.3, on the BA graphs, our model is about 2.3% away from optimal. The two machine learning baselines are slightly less than 1% away from optimal. The greedy baselines are $9\% - 45\%$ away from optimal.

**Qualitative Results.** Figure 5 shows typical results of our learned MVC heuristics. See §E.2 for more examples. On the ER graphs, we can see that the heuristic does not always start with the highest degree node. In contrast, on the BA graphs, the heuristic has a strong preference to start with the highest degree node. Unlike the classic greedy heuristics (and our learned graph coloring heuristic), the learned MVC heuristics seldomly pick neighboring nodes subsequently.

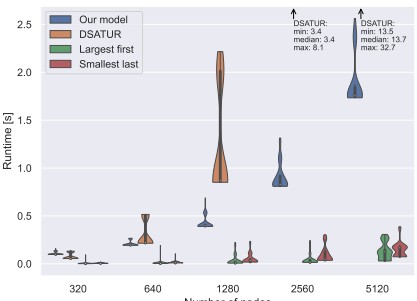

Figure 6: Distribution of graph coloring inference runtime on WS graphs.

Table 4: Mean validation cost of our approach for graph coloring on WS graphs and cost and approximation ratio for minimum vertex cover on ER graphs (10 samples). Shaded columns are for graphs larger than during training.

| | Nodes | 20 | 50 | 100 | 200 | 400 | 600 |
|---|---|---|---|---|---|---|---|
| GC | Cost | 3.92 | 4.00 | 4.02 | 4.01 | 4.04 | 4.05 |

| | Nodes | 15–20 | 40–100 | 100–300 | 300–500 | 500–600 |
|---|---|---|---|---|---|---|
| MVC | Cost | 8 | 43.8 | 178.4 | 384.9 | 540 |
| | Approx. | 1.012 | 1.042 | 1.048 | 1.055 | 1.054 |

**Runtime.** As shown in §B.5, our approach takes around 0.5 second to find a cover on the test graphs with 1,000 nodes on a CPU per sample, and around 5 seconds for 10 samples. This is comparable to what Dai et al. (2017) reported on a GPU on a similar graph (11 seconds). Note that the time budget for Li et al. (2018) was 30 seconds and the time budget for the combinatorial solver was 1 hour.

## 4.3 ABLATION STUDIES

**Spatial locality.** We test the inductive biases we made regarding locality of the decoder by comparing against a decoder variant that never updates the attention weights (*static decoding*) and a variant that always updates all of the attention weights (*global decoding*).

*Static decoding* never recomputes the attention weights. For node a node $i$ that is not yet labeled, its weight is: $a_i = C \cdot \tanh\left(\frac{(\Theta_1 \mathbf{g_o})^T (\Theta_2 h_i)}{\sqrt{d}}\right)$. Static decoding uses $\mathcal{O}(d^2 n + m + n^2)$ operations, which are fewer than those of local update decoding when $m \gg d^2 n$. With static decoding, the model is essentially a GNN with a special node-readout function.

*Global decoding* recomputes the attention weights in each time step $t$. For a node $i$ that is not yet labeled, its weight is: $a_i^{(t)} = C \cdot \tanh\left(\frac{(\Theta_1 \mathbf{g_t})^T (\Theta_2 h_i)}{\sqrt{d}}\right)$. Global decoding uses $\mathcal{O}(d^2 n^2)$ operations, which is at least a $d^2$ factor more than local update decoding for not too dense graphs ($m \ll n^2/d^2$). When there are only two labels (as for MVC), global decoding is very similar to the Kool et al. (2019) model. The difference to Kool et al. (2019) is that they use additional attention layer to compute a new state embedding in each step.

We train graph coloring models with both static and global decoding on the Lemos et al. (2019) subset of the COLOR challenge graphs (following §4.1). Static and global decoding achieve a mean cost of $10.74^{\pm 0.12}$ and $10.71^{\pm 0.05}$, respectively, both worse than when using our inductive bias (Table 1).

**Architecture Parameters.** We varied the size of the context embedding (i.e., the number of nodes and their labels that contribute to it). Increasing the context size does not significantly improve the test score on graph coloring. For graph coloring, a context of size two and three results in a mean cost of $10.49^{\pm 0.12}$ and $10.42^{\pm 0.12}$, respectively, for the greedy policy. We varied the number of attention heads (among 1, 2, 4) with a per-head dimension of 16. For graph coloring, this results in a mean validation cost of $5.29^{\pm 0.02}$, $4.98^{\pm 0.01}$, and $4.95^{\pm 0.02}$, respectively. We therefore use 4 attention heads (hidden dimension 64). In §B.4, we provide additional ablation studies for the encoder.

## 5 CONCLUSION

We introduced *combinatorial node labeling*, a framework that generalizes existing approaches to many hard graph problems, and presented a neural network architecture for it, which demonstrates excellent results on both graph coloring and minimum vertex cover problems. This serves as an important step toward replacing hand-crafted heuristics in graph algorithms with *learned* heuristics tailored to a particular problem and graph structure.

There are many avenues for future research. While the nodel labeling framework is very general, other graph problems may require adjustments to the neural architecture or inductive biases for good performance. In particular, handling weighted graphs and edge labeling problems would be valuable.

ETHICS STATEMENT

The combinatorial node labeling framework and our neural network architecture target a broad class of graph problems, and hence are very general-purpose. Downstream tasks of such problems range from compiler passes to logistics optimization to graph data mining. Hence, it is hard to identify specific cases of benefit or harm from our work, as it depends on the specific application of the downstream tasks. We nevertheless urge careful consideration of the implications of improving performance on tasks using our methods, especially ones with privacy implications (e.g., data mining).

REPRODUCIBILITY STATEMENT

We detail the combinatorial node labeling framework in §2, and describe the neural network architecture and training process we use in §3. We note that the node labeling framework is very general and alternative architectures could be used to solve it. Proofs of our theoretical claims are provided in §C and we give details of our training setup in §4 and §A. We additionally include our source code in the supplementary material to aid reproducibility.

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

# A  TRAINING

## A.1  DATA GENERATION

We use four different synthetic graph distributions to generate instances for training and validation. All graphs are generated via the Python NETWORKX library (Hagberg et al., 2008).

**Barabási-Albert Model (Albert & Barabási, 2002)**  The Barabási-Albert (BA) Model generates random scale-free networks. Similar to real-world networks BA graphs grow by preferential attachment, i.e., a new node is more likely to link to more connected nodes. The model is parameterized by one parameter $\delta$, which dictates the average degree.

**Erdős-Rényi Model (Erdős & Rényi, 1960)**  An Erdős-Rényi (ER) graph $G(n, p)$ has $n$ nodes and each edge exists independently with probability $p$. The expected number of edges is $\binom{n}{2}p$.

**Watts-Strogatz Model (Watts & Strogatz, 1998)**  Watts-Strogatz (WS) graphs were developed to overcome the shortcomings of ER graphs when modeling real world graphs. In real networks we see the formation of local clusters, i.e., the neighbors of a node are more likely to be neighbors. For parameters $k$ and $q$, a WS graph is built as follows: build a ring of $n$ nodes. Next, connect each node to its $k$ nearest neighbors. Finally, replace each edge $\{u, v\}$ by a new edge $\{u, w\}$ (chosen uniformly at random) with probability $q$.

### A.1.1  TRAINING SET PARAMETERS

See Table 5 for the parameters of the graph distributions used during training. Note that for BA and ER graphs, the parameters match those used in the Dai et al. (2017) test set (see Table 2 and Table 10). We also consider sparse ER graphs (S-ER), for we set the edge probability such that graphs have expected average degree close to $\Delta = 7.5$ when $n$ is small but remain connected with high probability when $n$ is large. This means that

$$p_{s-er} = \min\left(1, \max\left(\frac{\Delta}{n}, (1 + \epsilon)\frac{\ln n}{n}\right)\right) \ , \tag{1}$$

for a small $\epsilon$, which we set to $0.2$ in our experiments. The formula is derived from the connectivity threshold of ER graphs (Erdős & Rényi, 1960).

For graph coloring, we train on a hybrid dataset consisting of an equal proportion of BA, S-ER, and WS graphs. For minimum vertex cut, we train on a dataset consisting of BA graphs, a dataset consisting of ER graphs, and a hybrid dataset consisting on a combination of the two (in equal proportion). We use the in-distribution models for the evaluation on the synthetic test instances and the hybrid model for the `memetracker` graph. During training, we use an equal proportion of graphs with $n \in \{20, 40, 50, 70, 100\}$ nodes.

For the results on simple graphs in Table 3, we use graphs with sizes $n \in \{10, \dots, 51, 60, 61, 70, 71, 80, 81, 90, 91, 99, 100, 200, 201, 300, 301, 400, 401\}$ and validate on 1000 or 1001 nodes.

## A.2  POLICY OPTIMIZATION

We train our model with REINFORCE with a greedy rollout baseline Kool et al. (2019). The details follow. We denote the cost of labeling the graph $G_i$ in the order given by the sequence of nodes $\pi$ by $\mathcal{L}(\pi, G_i)$. A model $M$ is parameterized by parameters $\theta$. On a graph $G_i$, the model returns a sequence of nodes $\pi$ and an associated probability $p_\theta$. The probability $p_\theta$ is the product of all action probabilities that led to the sequence of nodes $\pi$. We write $p_\theta, \pi \leftarrow M_\theta(G_i)$ when the policy is evaluated deterministically and $p_\theta, \pi \sim M_\theta(G_i)$ when the policy is evaluated probabilistically.

Table 5: The graph parameters for training and validation.

| BA | ER | S-ER | WS |
|---|---|---|---|
| $\delta = 4$ | $p = 0.15$ | $p = p_{s-er}$ | $k = 5,$ $q = 0.1$ |

**Algorithm 1** Policy Training with Reinforce+Baseline

1: **Input:** number of epochs $E$, batch size $B$, datasets $D_{\text{train}}$
2: Initialize model $M_\theta$ and baseline model $M_\theta^{BL}$
3: $D_{\text{challenge}} \leftarrow$ sample new challenge dataset
4: **for** $epoch = 1, \ldots, E$ **do**
5:      **for** $batch$ in $D_{train}$ **do**
6:          $[ \ p_{\theta,i}, \pi_i \sim M_\theta(G_i) \ $ for $G_i$ in $batch \ ]$            $\triangleright$ Sample from policy
7:          $[ \ p_{\theta,i}^{BL}, \pi_i^{BL} \leftarrow M_{\theta^{BL}}(G_i) \ $ for $G_i$ in $batch \ ]$        $\triangleright$ Greedy baseline
8:          $\nabla_\theta J(\theta) = \frac{1}{B} \sum_{i=1}^{B} (\mathcal{L}(\pi_i \mid G_i) - \mathcal{L}(\pi_i^{BL} \mid G_i)) \ \nabla_\theta \log(p_{\theta,i})$    $\triangleright$ Policy Gradient
9:          $\theta \leftarrow \text{ADAM}(\theta, \nabla_\theta J(\theta))$
10:      **end for**
11:      **if** OneSidedPairedTTest$(M_\theta, M_\theta^{BL}, D_{\text{challenge}}) < 0.05$ **then**      $\triangleright$ Challenge the baseline
12:          $\theta^{BL} \leftarrow \theta$
13:          $D_{\text{challenge}} \leftarrow$ sample new challenge dataset
14:      **end if**
15: **end for**

The complete training procedure is given in Algorithm 1. Algorithm 1 follows from the textbook REINFORCE with a baseline (Sutton & Barto, 2018) by factoring the probability of reaching a terminal state and using that the rewards are 0 in our MDP except when reaching a terminal state. Unlike Kool et al. (2019), we do not use warmup epochs where training starts out with an exponential moving average baseline.

# B    ADDITIONAL RESULTS

## B.1    VALIDATION RESULTS

We compare the cost of the learned heuristic for different parameters of the training. The validation set consists of 600 graphs with $n$ nodes for $n \in \{20, 50, 100, 200, 400, 600\}$.

### B.1.1    GRAPH COLORING

Table 6 shows the validation cost on the three training distribution for all evaluated configurations.

With a larger learning rate of $\alpha = 10^{-3}$, the mean validation cost for graph coloring is significantly worse, namely $5.22^{\pm 0.002}$. A smaller learning rate of $\alpha = 10^{-5}$ leads to a mean validation cost of $5.02^{\pm 0.001}$, which is slightly worse than the cost of $4.95^{\pm 0.02}$ for $\alpha = 10^{-4}$.

### B.1.2    MINIMUM VERTEX COVER

Table 7 shows the validation results for training on either only one distribution and evaluating on ER and BA graphs. Training on a mixture ER and BA graphs leads to worse validation cost on BA graphs compared to training only on BA graphs. Training on ER graphs exclusively without BA graphs leads to a slight cost improvement on ER graphs.

## B.2    RESULTS BY SIZE

Table 8 shows how the cost and approximation ratio of our MVC approach varies with the instance size (on the ER test graphs). Although the approximation ratio grows slightly with instance size, it remains within ca 5.5% of optimal for graphs with 500-600 nodes.

Table 9 shows how the cost of GC varies on two synthetic distributions of graphs, S-ER and BA. For BA and ER graphs, the cost grows by about one color on the larger graphs.

## B.3    ADDITIONAL RESULTS FOR MVC

**Results on BA graphs**    See Table 10 for the MVC results on BA graphs.

Table 6: Ablation studies for graph coloring. *L*, *S*, and *G* are short for local, static, and global decoders, respectively. *Prev.* denotes the number of previously labeled nodes in the state embedding, *Norm.* indicates if batch-normalization is used, and *Short.* indicates if shortcuts are used.

| Encoder | | | Decoder | | | Validation cost on distribution | | | |
|---|---|---|---|---|---|---|---|---|---|
| Layers | Norm. | Short. | Type | Prev. | Heads | S-ER | WS | BA | All |
| 3 | ✓ | ✓ | L | 1 | 4 | **5.32**$^{\pm0.05}$ | **4.01**$^{\pm0.00}$ | **5.50**$^{\pm0.04}$ | **4.95**$^{\pm0.02}$ |
| **2** | ✓ | ✓ | L | 1 | 4 | 5.34$^{\pm0.04}$ | **4.01**$^{\pm0.01}$ | 5.57$^{\pm0.02}$ | 4.98$^{\pm0.02}$ |
| **1** | ✓ | ✓ | L | 1 | 4 | 5.4$^{\pm0.05}$ | 4.09$^{\pm0.01}$ | 5.55$^{\pm0.05}$ | 5.02$^{\pm0.03}$ |
| 3 | ✗ | ✗ | L | 1 | 4 | 5.56$^{\pm0.02}$ | 4.29$^{\pm\pm0.09}$ | 6.09$^{\pm0.04}$ | 5.32$^{\pm0.05}$ |
| 3 | ✓ | ✗ | L | 1 | 4 | 5.56$^{\pm0.05}$ | 4.12$^{\pm0.05}$ | 5.92$^{\pm0.04}$ | 5.22$^{\pm0.04}$ |
| 3 | ✓ | ✓ | **S** | 1 | 4 | 5.59$^{\pm0.04}$ | 4.14$^{\pm0.00}$ | 5.52$^{\pm0.06}$ | 5.09$^{\pm0.04}$ |
| 3 | ✓ | ✓ | **G** | 1 | 4 | 5.58$^{\pm0.02}$ | 4.15$^{\pm0.01}$ | 5.54$^{\pm0.03}$ | 5.1$^{\pm0.03}$ |
| 3 | ✓ | ✓ | L | **2** | 4 | 5.37$^{\pm0.03}$ | 4.02$^{\pm0.01}$ | **5.49**$^{\pm0.04}$ | 4.97$^{\pm0.03}$ |
| 3 | ✓ | ✓ | L | **3** | 4 | 5.36$^{\pm0.02}$ | 4.02$^{\pm0.01}$ | **5.48**$^{\pm0.07}$ | 4.96$^{\pm0.02}$ |
| 3 | ✓ | ✓ | L | 1 | **2** | 5.38$^{\pm0.03}$ | 4.02$^{\pm0.01}$ | 5.54$^{\pm0.03}$ | 4.98$^{\pm0.01}$ |
| 3 | ✓ | ✓ | L | 1 | **1** | 5.58$^{\pm0.02}$ | **4.01**$^{\pm0.01}$ | 6.26$^{\pm0.05}$ | 5.28$^{\pm0.02}$ |

Table 7: MVC Validation Cost for varying training and validation distributions.

| Train Graphs | Cost ER | Cost BA |
|---|---|---|
| ER | **223.56**$^{\pm0.11}$ | 218.07$^{\pm4.36}$ |
| BA | 223.76$^{\pm0.43}$ | **199.44**$^{\pm12.02}$ |
| ER+BA | 223.98$^{\pm0.26}$ | 208.18$^{\pm6.24}$ |

Table 8: Cost and approximation ratio of our approach for MVC on BA graphs by instance size. Shaded columns are for test instances larger than the training graphs. *Approx.* denotes the approximation ratio.

| Nodes | 15-20 | 40-100 | 100-300 | 300-500 | 500-600 |
|---|---|---|---|---|---|
| Cost | 10.53 | 35.08 | 113.56 | 225.16 | 309.02 |
| Approx. | 1.009 | 1.016 | 1.022 | 1.025 | 1.027 |

Table 9: Mean validation cost of our greedy approach for GC on S-ER and BA graphs by instance size. Shaded columns are for instances larger than those seen during training.

S-ER Graphs

| Nodes | 20 | 50 | 100 | 200 | 400 | 600 |
|---|---|---|---|---|---|---|
| Cost | 5.30 | 5.18 | 5.18 | 5.28 | 5.39 | 5.96 |

BA Graphs

| Nodes | 20 | 50 | 100 | 200 | 400 | 600 |
|---|---|---|---|---|---|---|
| Cost | 4.83 | 5.08 | 5.34 | 5.61 | 5.96 | 6.18 |

Table 10: Comparison of MVC approaches on BA graphs with average degree 4.

| | Name | Cost | Approx. Ratio |
|---|---|---|---|
| Class. | Maximal Matching | 190.84 | 1.4516 |
| | List Right | 143.50 | 1.0984 |
| ML | Li et al. (2018) | **131.62** | **1.0084** |
| | S2V-DQN | N/A | 1.0099 |
| | Ours - Greedy | 133.78$^{\pm0.07}$ | 1.0234 |
| | Ours - 10 Samples | 133.39$\pm$0.05 | 1.0202 |

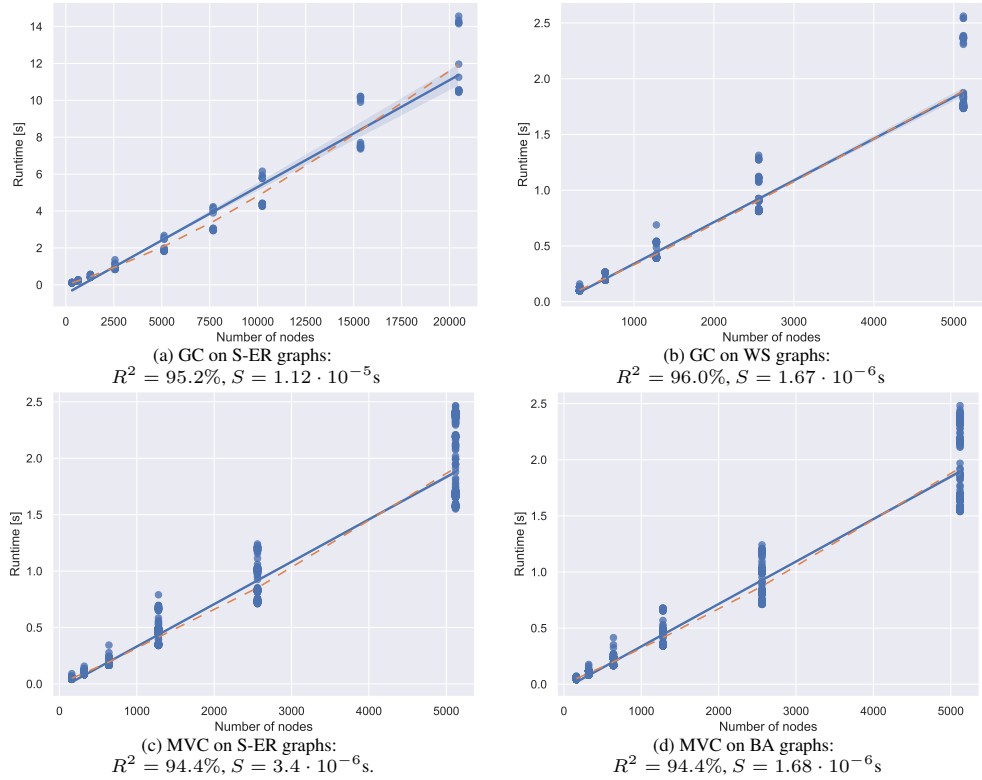

Figure 7: Inference runtime with local decoding. The solid line indicates a linear least squares fit, the dashed orange line the mean. We report the coefficient of determination $R^2$ and standard errror $S$.

## B.4 ABLATION STUDIES FOR THE ENCODER

We varied the number of layers in the encoder, removed the shortcut connections, and removed the normalization. The results are summarized in Table 6. We can see that removing the shortcut connections has a very strong detrimental effect on the validation cost. Removing both shortcuts and normalization deteriorates the cost further. Using 2 layers results in a small, but noticeable increase in cost, whereas a single layer has a significantly worse cost.

## B.5 RUNTIME SCALABILITY

We evaluated the inference runtime on a c2d-highmem-4 (4 vCPUs, 32 GB RAM) Google Cloud machine (Environment M94 with PyTorch 1.11). Figure 7 show the runtime scaling of our approach on GC and MVC together with a linear least squares fit. In Figure 7b, we see that for graphs up to around 5000 vertices the runtime of GC inference very closely follows a linear trend. For larger graphs, the runtime grows slightly faster than linear, as shown in Figure 7a. Similar results hold for MVC: it takes less than 0.5 seconds to compute a vertex cover for a graph with 1,000 nodes. Figure 7d and Figure 7c show the distribution of MVC runtimes of up to 5120 nodes.

## B.6 DISCUSSION.

It is not surprising that our machine learning approach performs best on in-distribution graphs. Whilst it is desirable to have an approach that generalizes well, if a representative sample of graphs is available for a target applications, this does not pose an issue. Moreover, we have shown that the quality degrades gradually when the test distribution differs from the training distribution.

## C  ADDITIONAL PROOFS

### C.1  THE NODE LABELING MDP

*Proof of Lemma 2.4.* Consider a sequence of actions $(v_1, \ell_1), \ldots, (v_n, \ell_n)$ ending in a terminal state. For all $t$, the prefix $(v_1, \ell_1), \ldots, (v_t, \ell_t)$ of this sequence corresponds to a partial node labeling $c'$ (by viewing the sequence of node-label pairs as describing a function from nodes to labels). By construction of the MDP, labeling node $v_{i+1}$ with $\ell_{t+1}$ passes the extensibility test for $c'$. Hence the node labeling $c$ represented by $(v_1, \ell_1), \ldots, (v_n, \ell_n)$ is feasible. By construction, the return of the episode is $-f(c)$, where $f(c)$ is the cost of node labeling $c$.

Conversely, consider a feasible solution $c$ with cost $f(c)$. Then, by definition of feasibility (§4.3), there is a sequence $(v_1, \ell_1), \ldots, (v_n, \ell_n)$ of node-label pairs such that for all $t \geq 0$ the partial node labeling given by $(v_1, \ell_1), \ldots, (v_t, \ell_t)$ passes the extensibility test for node $v_{t+1}$ and label $\ell_{t+1}$. Hence, the sequence of node-label pairs is also a sequence of actions in the MDP leading to a terminal state. The return for this episode is $-f(c)$.

Note that since our tasks are episodic, the return equals the sum of the rewards (specifically the reward received in the terminal state). In particular, we do not use discounting.

□

### C.2  OPTIMALITY OF THE LABELING RULE

*Proof of Lemma 3.1.* Let $G$ be some graph with chromatic number $\chi(G) = k$ and $c^*$ be a mapping that colors $G$ optimally. We partition $V$ into color classes $C_i = \{v \mid c^*(v) = i\}$ such that all nodes with color $i$ are in $C_i$. Now, we build an ordering by consecutively taking all nodes from $C_1$, then all nodes from $C_2$ and so on. Choosing the smallest color that passes the extensibility test will produce an optimal coloring for such an order of nodes: The proof is by strong induction on the index of the color class $i$. The induction hypothesis $H(i)$ is that for all nodes $v$ in $C_j$ for $j < i$, $v$ is colored with a color in $\{1, \ldots, j\}$. Assume the induction hypothesis $H(i)$ holds. Now, consider a node $v$ in $C_i$. The color $i$ must be a valid color for $v$: First, assigning color $i$ does not produce any conflicts with any node $u$ in $C_j$ for $j < i$ because by induction hypothesis node $u$ has a color strictly less than $i$. Second, assigning color $i$ to $v$ does not produce a conflict with another node $w$ in $C_i$ because then $C_i$ would not be a valid class of colors (nodes in a color class cannot be neighbors.). As we choose the smallest valid color and $i$ is valid, $v$ get a color in $\{1, \ldots i\}$. Thus, $H(i+1)$ holds.

Note that this coloring might be different from the one of $c^*$. This is, because a node in $C_i$ might have no conflicts with some color $j < i$ and therefore this node will be assigned color $j$. □

*Proof of Lemma 3.2.* Let $S$ be the set over nodes with label 1 in a minimum vertex cover of $G$. Order these nodes first (in an arbitrary relative order), then order the remaining nodes in $V - S$ after these nodes (in an arbitrary relative order). Now, label the nodes with 1 in this order until every edge has an endpoint with label 1. After $|S|$ steps, every node in $S$ has label 1, meaning that the nodes with label 1 form a minimum vertex cover: If the nodes formed a vertex cover after less than $|S|$ steps, we would find a smaller vertex cover, contradicting the minimality of $S$. □

## D  LIST OF COMBINATORIAL NODE LABELING PROBLEMS

We provide an extensive list of classic graph optimization problems framed as node labeling problems. Note that there can be multiple equivalent formulations. For some problems, we consider a *weighed graph* $G$ with weight function $w : E \mapsto \mathbb{R}^+$, we write $w(u, v)$ the weight of an edge $\{u, v\}$. For a set of nodes $S$, we denote the subgraph of $G$ induced by $S$ with $G[S]$.

The problems in Table 11 require a partition of the nodes as their solution. These can be represented as node labeling problems by giving each partition its unique label. For many of the problems, the number of used labels determines the cost function.

The problems in Table 12 require a path (or a sequences of nodes) as their solution, which we represent as node labeling problems by having the label indicate the position in the path (or sequence).

Table 11: Node labeling problems which partition the nodes into 2 or more sets.

| Problem | Extensibility Test $T(V' \times \mathcal{L}, v, l)$ | Cost function $f$ |
|---|---|---|
| Balanced $k$-partition (Kernighan & Lin, 1970) | There are no more than $\lceil \frac{n}{k} \rceil$ nodes with the same label and at most $k$ labels. | $\sum_{\{u,v\}\in E, l(u)\neq l(v)} w(u,v)$ |
| Balanced $k, 1+\epsilon$ vpartition (Kernighan & Lin, 1970) | There are no more than $\lceil \frac{n(1+\epsilon)}{k} \rceil$ nodes with the same label and at most $k$ labels. | $\sum_{\{u,v\}\in E, l(u)\neq l(v)} w(u,v)$ |
| Minimum $k$-cut (Karger & Stein, 1996) | $k - |V| - |V'| - 1 \leq |\mathcal{L} \cup \{v\}|$ and $|\mathcal{L} \cup \{v\}| \leq k$ | $\sum_{\{u,v\}\in E, l(u)\neq l(v)} w(u,v)$ |
| Clique cover (Karp, 1972) | Every label induces a clique | Number of labels |
| Domatic number (Hedetniemi & Laskar, 1990) | Every label induces a dominating set of $G[V' \cup \{v\}]$ | Negative number of labels |
| Graph coloring (Jensen et al., 1995) | No neighbor of $v$ has label $l$ | Number of labels |
| Graph co-coloring (Jensen et al., 1995) | The nodes with label $l$ induce an independent set in $G$ or the complement of $G$ | Number of labels |
| $k$-defective coloring (Cowen et al., 1986) | No node has more than $k$ neighbors with label $l$ | Number of labels |

Table 12: Node labeling problems where the labels encode a permutation of nodes.

| Problem | Extensibility Test $T(V' \times \mathcal{L}, v, l)$ | Cost function $f$ |
|---|---|---|
| Traveling salesman problem (Dantzig et al., 1954) | $l = \max(\mathcal{L}) + 1$ and $v$ is a neighbor of the node in $\mathcal{L}$ with label $\max(\mathcal{L})$ | $\sum_{(u,v)\in E, l(v)=l(u)+1} w(u,v)$ |
| Tree decomposition (Bodlaender, 2005) | $l = \max(\mathcal{L}) + 1$ | For a node $v_i$ with label $i$, add edges to $G$ until $v_i$ forms a clique with its higher-labeled neighbors. The cost is the largest number of higher-labeled neighbors in the augmented graph (Bodlaender, 2005). |
| Longest path (Karger et al., 1997) | $l = \max(\mathcal{L}) + 1$ | Maximum number of nodes with consecutive labels that induce a path |

The problems in Table 13 require a set of nodes as their solution. These can be represented as node labeling problems by giving the nodes in the solution set the label 1 and the nodes not in the solution set the label 0. The cost function is closely related to the number of nodes with label 1 for most of these problems.

Table 13: Node labeling problems with binary labels. *For all these problems*, the extensibility test passes only if the label is 0 or 1 (and the additional requirements listed below are satisfied).

| Problem | Extensibility Test $T(V' \times \mathcal{L}, v, l)$ | Cost function $f$ |
|---|---|---|
| Maximum cut (Karp, 1972) | At least one node has label 1 | $-|\{\{u, v\} \in E, l(u) \neq l(v)\}|$ |
| Sparsest cut (Arora et al., 2009) | At least one node has label 1 | $\frac{|\{\{u,v\}\in E,\, l(u)\neq l(v)\}|}{|\{v\in V,\, l(v)=1\}|}$ |
| Maximum independent set (Tarjan & Trojanowski, 1977) (Robson, 1986) | The subgraph induced by the nodes with label 1 is an independent set | $-|\{v \in V, l(v) = 1\}|$ |
| Minimum vertex cover (node cover) (Karp, 1972) | The subgraph induced by the nodes with label 1 is a vertex cover of $G[V' \cup \{v\}]$ | $|\{v \in V, l(v) = 1\}|$ |
| Maximum clique (Tomita & Seki, 2003) | The subgraph induced by the nodes with label 1 is a clique | $-|\{v \in V, l(v) = 1\}|$ |
| Minimum feedback node set (Karp, 1972) | $G[\{u \in V' \cup \{v\}, l(u) = 0\}]$ is a forest | $|\{v \in V, l(v) = 1\}|$ |
| Metric dimension (Harary & Melter, 1976) | The nodes in $V' \cup \{v\}$ are uniquely identified by their distances to nodes with label 1 | $|\{v \in V, l(v) = 1\}|$ |
| Minimum dominating set (Hedetniemi & Laskar, 1990) | The nodes with label 1 form a dominating set of $G[V' \cup \{v\}]$ | $|\{v \in V, l(v) = 1\}|$ |
| Minimum connected dominating set (Hedetniemi & Laskar, 1990) | The nodes with label 1 form a connected dominating set of $G[V' \cup \{v\}]$ | $|\{v \in V, l(v) = 1\}|$ |

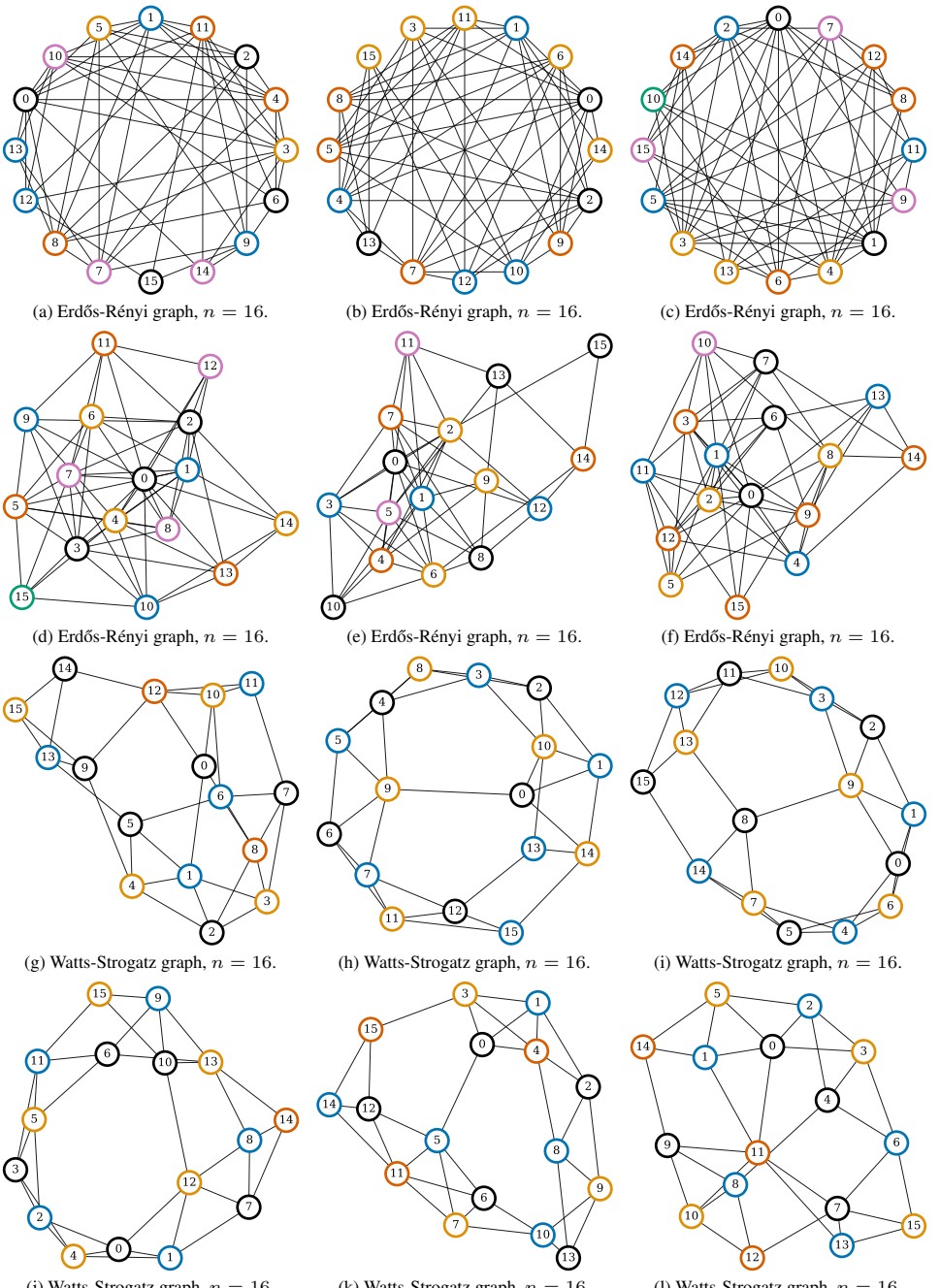

(a) Erdős-Rényi graph, $n = 16$.  (b) Erdős-Rényi graph, $n = 16$.  (c) Erdős-Rényi graph, $n = 16$.

(d) Erdős-Rényi graph, $n = 16$.  (e) Erdős-Rényi graph, $n = 16$.  (f) Erdős-Rényi graph, $n = 16$.

(g) Watts-Strogatz graph, $n = 16$.  (h) Watts-Strogatz graph, $n = 16$.  (i) Watts-Strogatz graph, $n = 16$.

(j) Watts-Strogatz graph, $n = 16$.  (k) Watts-Strogatz graph, $n = 16$.  (l) Watts-Strogatz graph, $n = 16$.

Figure 8: Example colorings produced by our learned heuristic. Node borders indicate the colors. Numbers on the nodes indicate the order in which the heuristic labels them.

# E  ADDITIONAL EXAMPLES

## E.1  GRAPH COLORING

Figure 8 and Figure 9 show additional results of our learned coloring heuristic on in-distribution graphs. For the Erdős-Rényi graphs and Watts-Strogatz, we provide some examples in a circular layout and some with a force-directed layout. The force-directed layout emphasizes the structure of the graph, but for these two graph classes leads to many crossing edges. See Figure 10 and Figure 11 for results on cycles, wheels, and trees. Interestingly, the heuristic picks the highest degree node in a star or wheel sometimes first and sometimes last.

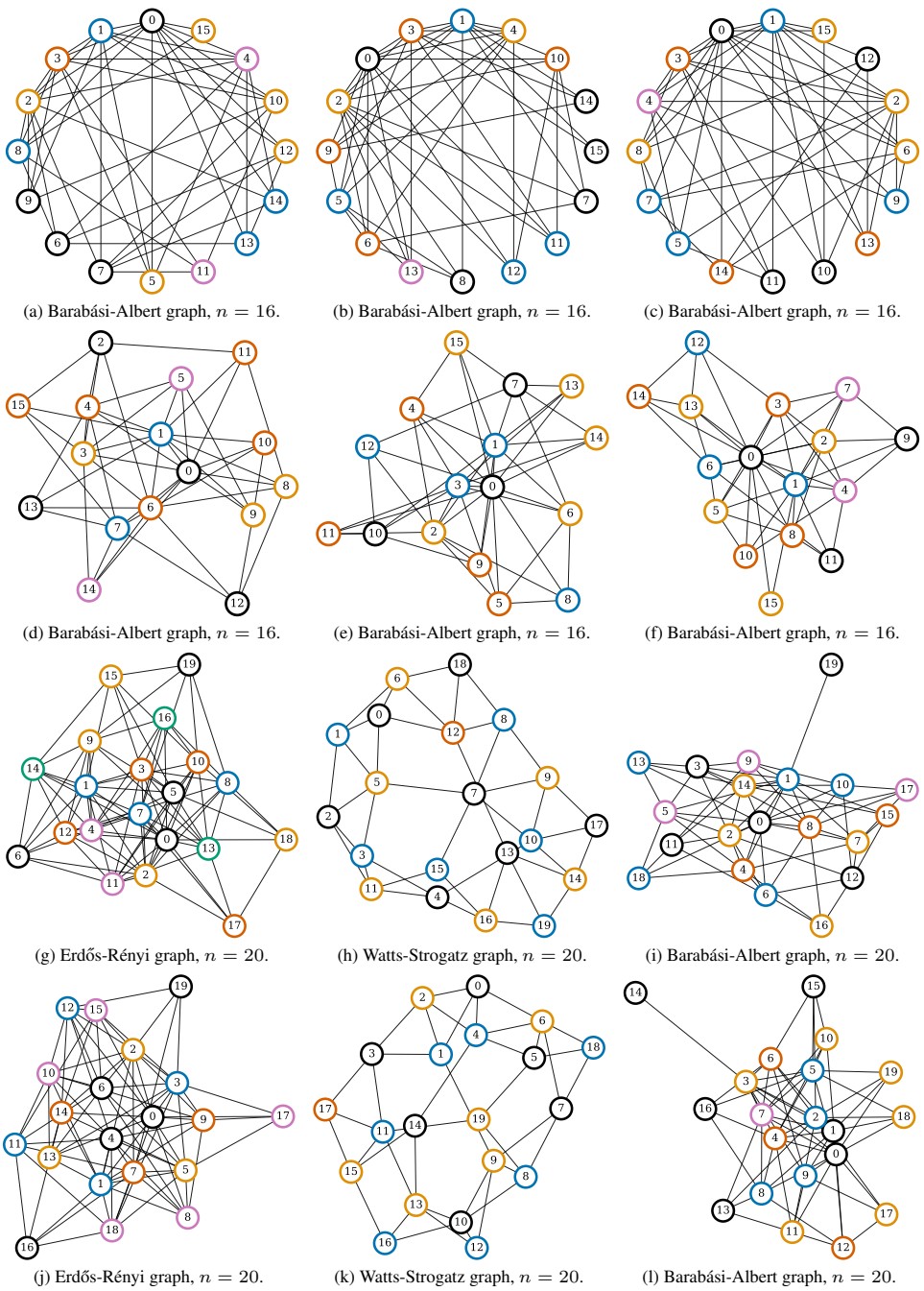

Figure 9: Example colorings produced by our learned heuristic. Node borders indicate the colors. Numbers on the nodes indicate the order in which the heuristic labels them.

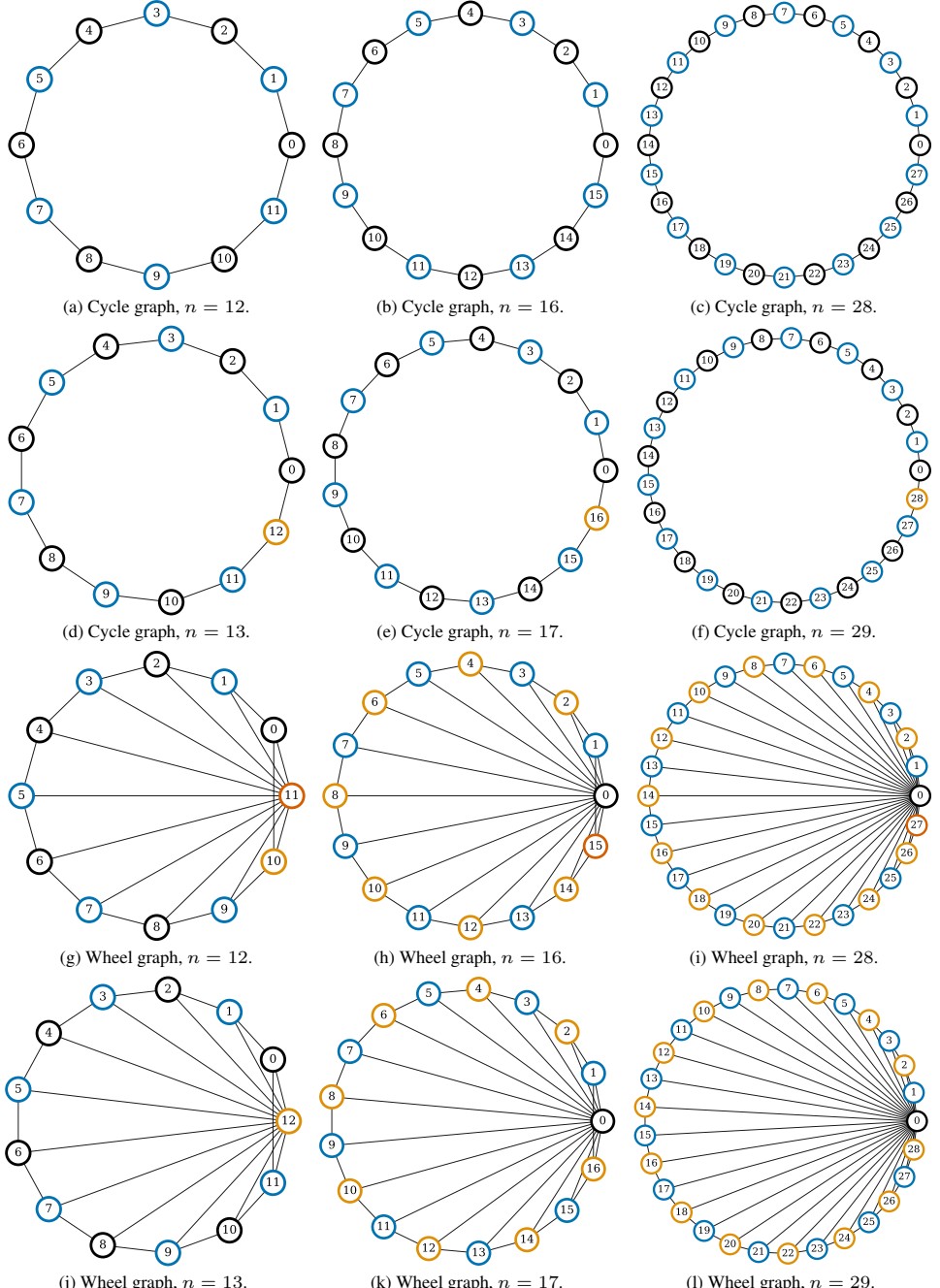

Figure 10: Example colorings produced by our learned heuristic on cycles and wheels. The learned coloring heuristic visits nodes on the cycles in-order. For the wheels, the center of the wheel is either visited first or last.

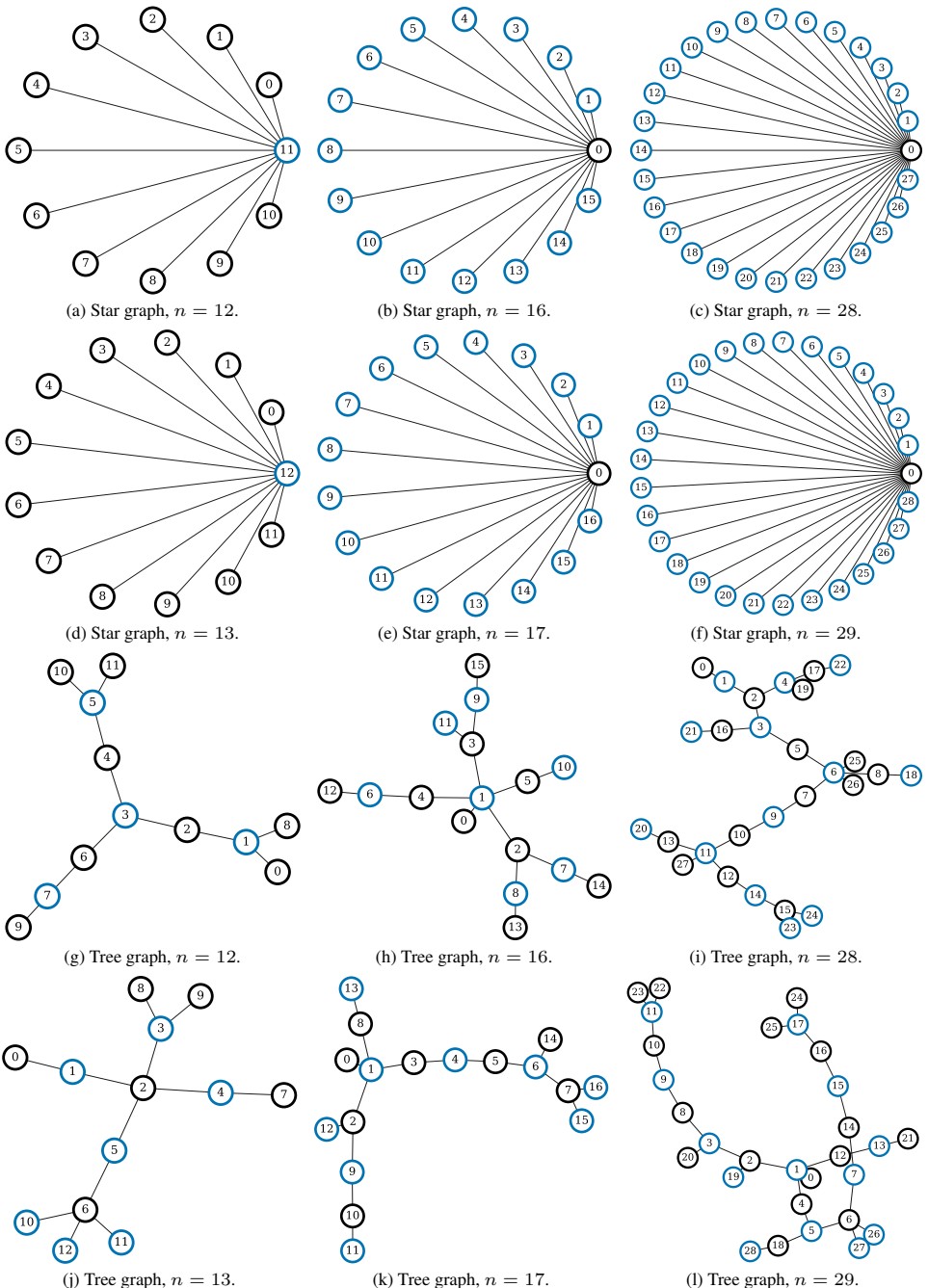

Figure 11: Example colorings produced by our learned heuristic on stars and random trees. For stars, the heuristic either labels the center first or last. The tree heuristic prefers to start coloring at one of the leaves of the tree and then colors nodes in a search pattern from there, coloring nodes that are neighbors of already colored nodes. It often labels the remaining leaves very late into the coloring.

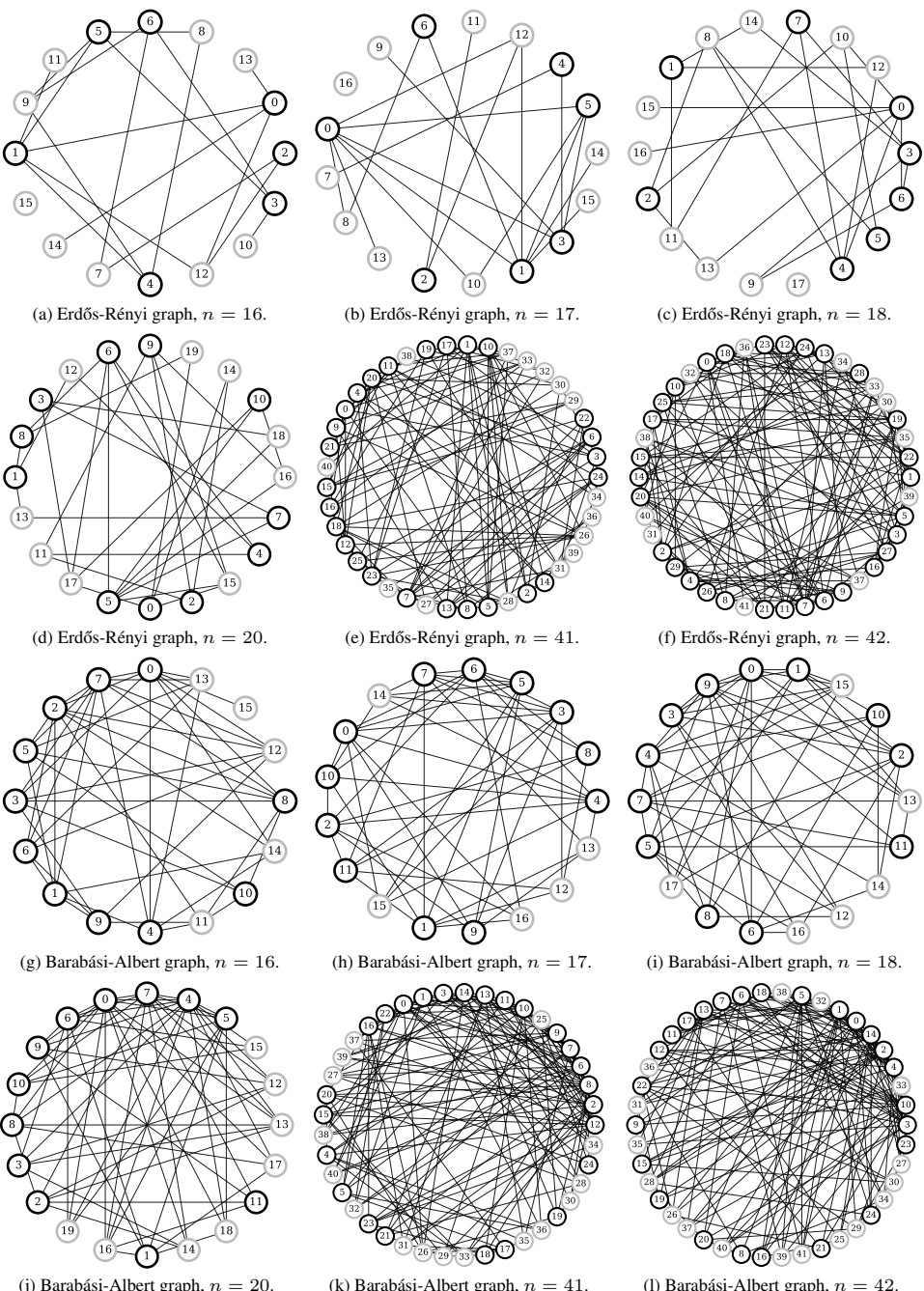

(a) Erdős-Rényi graph, $n = 16$.  (b) Erdős-Rényi graph, $n = 17$.  (c) Erdős-Rényi graph, $n = 18$.

(d) Erdős-Rényi graph, $n = 20$.  (e) Erdős-Rényi graph, $n = 41$.  (f) Erdős-Rényi graph, $n = 42$.

(g) Barabási-Albert graph, $n = 16$.  (h) Barabási-Albert graph, $n = 17$.  (i) Barabási-Albert graph, $n = 18$.

(j) Barabási-Albert graph, $n = 20$.  (k) Barabási-Albert graph, $n = 41$.  (l) Barabási-Albert graph, $n = 42$.

Figure 12: Example covers produced by our learned heuristic on Erdős-Rényi and Barabási-Albert graphs. Black-bordered nodes are in the cover.

## E.2  MINIMUM VERTEX COVER

Figure 12 shows additional example covers of our learned heuristic on in-distribution graphs.

