# OpenReview forum: "Learning Combinatorial Node Labeling Algorithms"
_ICLR.cc/2023/Conference — Submitted to ICLR 2023_

### Official Review · Reviewer_j1zE · 2022-10-23

**Confidence:** 4
**Correctness:** 3
**Technical Novelty And Significance:** 3
**Empirical Novelty And Significance:** 3
**Recommendation:** 5

**Clarity, Quality, Novelty And Reproducibility:**

The paper is presented in good clarity and quality. The idea is novel and the results are believed to be reproducible.

**Strength And Weaknesses:**

Strength
- The perspective of unifying CO problems in the framework of node labeling is novel and makes sense.
- Writing is good and the paper is easy to follow.
- Experiments are sufficient. Experimental settings (baselines, datasets, paremeter settings, etc.) are presented clearly, and results are convincing.
Weakness
- One of the biggest concerns is whether the framework really works on the task MVC, which is usually not processed as  the node labeling problem in the main stream of literature. And in this sense, baselines are too old. To make the results more convincing, more recent baselines, e.g. [1][2], and traditional solvers should be compared. Only by comparing the sota methods and traditional ones on both running time and the results on the MVC problem, can the paper prove that node labeling works in tasks beyond graph coloring. Whether it `really' works well on MVC is the key point to judge whether the significance of unifying the CO problems into the node labeling framework is over-claimed.
-The other concern is the technical novelty of the paper, it seems to me an increment extension from binary classification to the multi-class case based on existing S2V-DQN framework.

[1] Local Search with Efficient Automatic Configuration for Minimum Vertex Cover
[2] Efficient Minimum Weight Vertex Cover Heuristics Using Graph Neural Networks

**Summary Of The Paper:**

This paper proposes a combinatorial node labeling framework, which solves CO problems by labeling the nodes. The model is based on graph attention networks and trained by reinforcement learning. Graph coloring and minimum vertex cover are studied. Experiments prove the performance of the proposed framework.

**Summary Of The Review:**

This paper unifies a lot of CO problems into the node labeling framework. Experiments on graph coloring and MVC show the significance. The problem is that the significance of dealing CO problems as node labeling might be overclaimed. More experiments on MVC compare with stronger baselines or experiments on other tasks are recommended.

---

> ### Author Response · Authors · 2022-11-18
> **Response and Discussion**
>
> We thank the reviewer for their feedback.
>
> To address the mentioned concerns, we implemented the following revision:
>
> * We have added an additional comparison to the MVC results comparing them to two "classical" algorithms. The ML approaches generally outperform these baselines, except for S2V-DQN, which has a worse approximation ratio on ER graphs.
>
> Moreover, we would like to emphasize that the differences between Li. et al.'s results on MVC and ours are only around 2 percentage points in terms of optimality ratio.

---

### Official Review · Reviewer_aTgv · 2022-10-24

**Confidence:** 4
**Correctness:** 3
**Technical Novelty And Significance:** 2
**Empirical Novelty And Significance:** 3
**Recommendation:** 3

**Clarity, Quality, Novelty And Reproducibility:**

The authors include code in the supplementary so the clarity and reproducibility are good.

**Strength And Weaknesses:**

**Strength**
1. This paper extends the classic S2V-DQN approach, with the purpose to fit into more general problems.
2. Learning node labels is an interesting characteristic of learning CO, which may inspire future papers.

**Weakness**
1. The overall pipeline of this proposed approach is very similar to S2V-DQN, and such a reinforcement learning pipeline is actually followed by many other papers to tackle different problems since the S2V-DQN paper was published in 2017. For example, the following papers also study graph coloring:
    * Enhancing column generation by a machine-learning-based pricing heuristic for graph coloring. AAAI 2022.
    * Approximation ratios of graph neural networks for combinatorial problems. NeurIPS 2019.

   Considering these existing papers, the node-assignment step, which is the main technical contribution of this paper, seems incremental.

2. Adopting attention models for CO is either a new idea, for example, the highly cited paper:
    * Attention, learn to solve routing problems! ICLR 2018.

   So in this paper, replacing S2V with GAT in the model does not seem novel either.
2. Since the proposed scheme also covers "set" and "permutation" problems, it will be interesting to see the performance of the proposed method on well-studied problems such as TSP. It will be easier to position the effectiveness of this approach w.r.t. existing papers with the results on problems like TSP.

**Summary Of The Paper:**

This paper extends S2V-DQN and enables it to cover more node-labeling-related problems. In terms of the neural network model, S2V embedding is replaced by more recent GAT embedding. Experiments are conducted on graph coloring and minimum vertex cover.

**Summary Of The Review:**

The idea of "learning node labeling" for CO is novel and interesting, however, the technical contributions of this paper seem trivial and incremental. I am suggesting a rejection in consideration of the lack of technical novelty.

---

### Official Review · Reviewer_isah · 2022-10-24

**Confidence:** 4
**Correctness:** 2
**Technical Novelty And Significance:** 2
**Empirical Novelty And Significance:** 2
**Recommendation:** 3

**Clarity, Quality, Novelty And Reproducibility:**

The authors add their code in the supplementary material, to ensure reproducibility. However I did not test it.

For the reasons explained in the previous section, I do not think the quality of the paper is good enough.

I list here some minor points / typos:
- Section 2.1: a node labelling is a mapping V→{0,…,n} . n, the max label, could be different from the number of nodes. Coloring is an example of that.
- Lemma 3.2: “...choosing the label 1 until every vertex in G is adjacent to a node…” replace vertex with edge
- Section 3.2, paragraph Local attention decoder: in the definition of a_v^{(t)} change h_i with h_v
- Table 9 only shows cost for ER and BA graphs, while in the caption also WS graphs are mentioned

**Strength And Weaknesses:**


The idea to create an efficient way for node labeling in combinatorial optimization problems is new and promising. Indeed, in greedy algorithms (GA) one decides the rules to choose the next node to label. In GA these rules are chosen a-priori, and different choices are possible: taking inspiration from GA, the authors try to infer which is the best order using a Graph attention network: the resulting order for node labeling is different from the standard GA ones.

The main weakness of the paper is that the performances of the method are not extensively investigated and the comparison with existent methods to solve combinatorial optimization problems is not completely fair and exhaustive. Moreover, the weaknesses of the results are only shown in the Appendix and not in the main text.

For the graph coloring (GC), the authors compare their results with GAs and with the machine learning (ML) approach of Lemos et al (2019) (GNN-GCP). GNN-GCP is not very good for GC. In fact it acts as a classifier, with the only output being the predicted chromatic number and not a possible configuration of nodes labels. For this reason GNN-GCP can find a number of colors smaller than the correct one, that is a highly undesirable feature. I suggest to compare the results of the method proposed by the authors with other better ML approaches to GC, such as the one in Jan Toenshoff, Martin Ritzert, Hinrikus Wolf, and Martin Grohe. Frontiers in artificial intelligence, 3:98, 2021 (the arXiv reference in the draft should be updated with the published one) or the recent Schuetz, Martin JA, et al. "Graph Coloring with Physics-Inspired Graph Neural Networks." arXiv preprint arXiv:2202.01606 (2022).

For GC on simple family of graphs, like wheels, trees, even/odd cycles, after a training on graphs up to 400 nodes, the proposed method finds the good chromatic number on graphs up to 1000. The author could test the performances on larger instances in this very basic examples to check if performances degrade going to larger sizes.

The runtime is shown in Fig. 6-7 for GC and Fig. 8 for MVC. In this last case, the scaling seems to be exponential with the size of the problem and not linear. However in both cases a fit should be performed to show if the scaling is linear or not. (In Fig. 6 the scaling of DSATUR seems to be not-linear too; however DSATUR is a simple greedy method that should perform in linear time, thus I have some doubt on the good implementation of it).

When tested on sparse Erdos-Renyi graphs of average connectivity c=7.5, the method proposed by the authors finds a chromatic number of almost 5 for instances with 100 nodes, and a chromatic number of almost 6 for instances with 600 nodes. However a standard message passing algorithm, or a stochastic walk-sat algorithm can find a good coloring with 4 colors up to connectivity larger than 8 for much larger graphs in linear time (see for example L Zdeborova and F Krzakala. Phase transitions in the coloring of random graphs. Physical Review E, 76(3):031131, 2007). Thus the performances of the proposed method are really poor in this case and degrade fast with growing sizes.

For MVC, following table 10 the proposed method cannot reach results obtained by other existing ML methods. Moreover no comparisons with standard algorithms is performed.

In Table 7 one notices that performances of the proposed method quite strongly depend on the training set: if trained on ER graphs, the performances on BA graphs reduce and vice-versa: the method cannot generalize well.

**Summary Of The Paper:**

The authors present a graph attention network trained using policy gradient reinforcement learning that can label nodes in an efficient way to (try to) solve hard combinatorial optimization problems. They test their method on the graph coloring and minimum vertex cover problem on different kinds of graphs , comparing the performances with greedy algorithms and different neural networks approaches proposed in the past.

**Summary Of The Review:**

Summarizing while the idea of finding an efficient way for node labeling in hard combinatorial problems is new and interesting, the results obtained in the draft are not good enough: the proposed method cannot reach the performances of standard algorithms, it does not generalize well and its performances degrade going to larger sizes of the problems.

---

> ### Author Response · Authors · 2022-11-18
> **Revisions and Discussion**
>
> We thank the reviewer for their insightful feedback and suggestions.
>
> To address the mentioned concerns, we implemented the following revisions:
>
> * We extended the evaluation of GC on the "simple" families of graphs from Table 3 to graphs of up to 10000 nodes. The approach still works equally well on those larger graphs.
>
> * Note that the data in the runtime plots from Figure 6 and 8 were on a log-x axis. This would make a linear trend look exponential. We have changed the figures in the appendix to have a linear axis to improve their readability. Moreover, we performed a linear regression on the runtimes and report the results in the appendix (new Figures 7). Overall, the runtime is explained well by the linear fit. Note that the runtimes of MVC are improved compared to the first draft as we improved the speed of the cost computation which was a bottleneck for larger graphs.
>
> * We added a comparison with two greedy MVC algorithms that give provoble approximation guarantees. Our approach and Li's approach outperfroms them by a signficant margin on both ER and BA graphs. S2V-DQN is outperformed by one of the greedy heuristics (List Right) on ER graphs, but also beats both greedy baselines on BA graphs.
>
> * We clarified the definition of the node labeling and the partial node labeling to make it clear that the mapping is onto a subset of {0, ..., n}. We also fixed the other minor clarity issues.
>
> Moreover, we would like to clarify and discuss the following:
>
> * We used the DSATUR implementation from networkx. It is in fact possible to implement DSATUR in linear time and thus the observed scaling should be taken as a property of the implementation and we believe a tuned implementation of DSATUR could be comparable in speed with the other greedy baselines such as smallest last.
>
> * We do observe that in-distribution results are better than out-of-distribution results. We acknowledge that it would be desirable to have as good generalization as possible. However, when a good sample of the application graphs are available for training, we do not expect this to be a major issue. Fine-tuning could be a solution for the case where only a small number of application domain graphs are available.

---

### Official Review · Reviewer_wJkv · 2022-10-24

**Confidence:** 3
**Correctness:** 3
**Technical Novelty And Significance:** 2
**Empirical Novelty And Significance:** 3
**Recommendation:** 5

**Clarity, Quality, Novelty And Reproducibility:**

The paper is very well written and easy to follow. The entire training procedure, including hyperparams, is described, which results in transparency and good reproducibility. The paper is not very different from what the ML community has been doing in combinatorial optimization, so novelty is a bit limited (although it definitely exists).

**Strength And Weaknesses:**

--- Strength ---
a) The node labeling framework is a clear formulation of a wide class of problems and will help guide future contributions.
b) The graph coloring problem indeed never had, to the best of my knowledge, a more general ML solution.
c) Their MDP formulation allows for more than two labels and, more importantly, unknown number of labels. This is because a state is not a partial solution, but a partial labeling, which is fundamentally different.
d) Lemma 2.4 guarantees a solution in the MDP space, which is not a common result in previous literature.
e) Empirical results, especially for GC, are very promising.

--- Weaknesses ---
a) Throughout the text, the authors argue that node labeling and greedy strategies are intimately related (due to the ordering of labels). I understand that it's a natural relationship when we design heuristics, but we know that relationship does not necessarily exist in optimal solutions. I think it would be good to make this distinction more clear.
b) I find the proofs not rigorous enough (although the statements seem to be true to me). The authors try to present constructive arguments, which can result incur easily in this (without enough care). From what I see the proofs can be rewritten to use contradiction instead. A good example of this is Lemma 3.2's proof: The authors construct a solution and immediately conclude " Labeling the nodes in this order produces a minimum vertex cover of G.". I understand the lemma stands, but it reads sloppy.
c) I expected to see larger graphs in the dataset. Also, for ML in combinatorial opt, it is vital to test your solution on graphs that are larger at test time (shift in distribution).

**Summary Of The Paper:**

This work proposes a framework for learning combinatorial problems over graphs in which the solution can be expressed in terms of an ordered labeling of the nodes. Then, the authors propose GAT-CNL, a neural architecture leveraging the framework to learn to solve such class of problems in a greedy fashion. Empirically, the method presents better results when solving graph coloring and minimum vertex cover.

**Summary Of The Review:**

In my opinion the paper needs a little bit more of work, since the weaknesses currently outweigh the contributions. If the authors are able to test the solution on larger graphs and make the proofs more rigorous, I'm willing to raise my score.

---

> ### Author Response · Authors · 2022-11-18
> **Addressing the concerns**
>
> We thank the reviewer for their feedback and suggestions.
>
> To address the mentioned concerns, we implemented the following revisions:
>
> a) We clarified at the beginning of Section 3 that greedy solutions generally trade optimality for improved runtime and that our work falls into these category of works.
>
> b) We clarified the proofs of Lemma 3.1 and Lemma 3.2. Lemma 3.1 now makes an explicit (strong) induction argument. Both proofs now contain some arguments by contradiction. If another proof remains unclear, we would be happy to elaborate.
>
> c)  Moreover, Table 3 contains results on graphs that are more than 20 times larger than seen during training.
>
> Moreover we would like to point out that:
>
> c) Table 4 shows the behavior on graphs that are up to 6 times larger than the training distribution. Moreover, note that the test graphs from Table 1 do not come from the training distribution(s), but are challenge instances.

---

### Decision · Program_Chairs · 2023-01-20

**Decision:**

Reject

**Justification For Why Not Higher Score:**

The proposed method compares poorly to existing methods.

**Justification For Why Not Lower Score:**

N/A

**Metareview: Summary, Strengths And Weaknesses:**

The reviewers agreed that this paper has important drawbacks. I think the reviews summarize well the strengths and weaknesses of the paper. The proposed method compares poorly to existing methods.